# Feature blindness: A challenge for understanding and modelling visual object recognition

**Gaurav Malhotra**\*, **Marin Dujmović**, **Jeffrey S. Bowers**

School of Psychological Sciences, University of Bristol, Bristol, United Kingdom

\* gaurav.malhotra@bristol.ac.uk

## Abstract

Humans rely heavily on the shape of objects to recognise them. Recently, it has been argued that Convolutional Neural Networks (CNNs) can also show a shape-bias, provided their learning environment contains this bias. This has led to the proposal that CNNs provide good mechanistic models of shape-bias and, more generally, human visual processing. However, it is also possible that humans and CNNs show a shape-bias for very different reasons, namely, shape-bias in humans may be a consequence of architectural and cognitive constraints whereas CNNs show a shape-bias as a consequence of learning the statistics of the environment. We investigated this question by exploring shape-bias in humans and CNNs when they learn in a novel environment. We observed that, in this new environment, humans (i) focused on shape and overlooked many non-shape features, even when non-shape features were more diagnostic, (ii) learned based on only one out of multiple predictive features, and (iii) failed to learn when global features, such as shape, were absent. This behaviour contrasted with the predictions of a statistical inference model with no priors, showing the strong role that shape-bias plays in human feature selection. It also contrasted with CNNs that (i) preferred to categorise objects based on non-shape features, and (ii) increased reliance on these non-shape features as they became more predictive. This was the case even when the CNN was pre-trained to have a shape-bias and the convolutional backbone was frozen. These results suggest that shape-bias has a different source in humans and CNNs: while learning in CNNs is driven by the statistical properties of the environment, humans are highly constrained by their previous biases, which suggests that cognitive constraints play a key role in how humans learn to recognise novel objects.

## Author summary

Any object consists of hundreds of visual features that can be used to recognise it. How do humans select which feature to use? Do we always choose features that are best at predicting the object? In a series of experiments using carefully designed stimuli, we find that humans frequently ignore many features that are clearly visible and highly predictive. This behaviour is statistically inefficient and we show that it contrasts with statistical inference

**Data Availability Statement:** Data and computational code related to the paper are available at: https://github.com/feature-blindness/patchwork.

**Funding:** This research was funded by H2020 European Research Council Grant 741134 to JSB. The funders had no role in study design, data collection and analysis, decision to publish, or preparation of the manuscript.

**Competing interests:** The authors have declared that no competing interests exist.

models such as state-of-the-art neural networks. Unlike humans, these models learn to rely on the most predictive feature when trained on the same data. We argue that the reason underlying human behaviour may be a bias to look for features that are less hungry for cognitive resources and generalise better to novel instances. Models that incorporate cognitive constraints may not only allow us to better understand human vision but also help us develop machine learning models that are more robust to changes in incidental features of objects.

## Introduction

Sometimes we fail to see what's right in front of our eyes.

The seemingly simple task of recognising an object requires contending with a multitude of problems. Humans can recognise something as a "chair" for a vast range of lighting conditions, distances to the retina, viewing angles and contexts. We can recognise chairs made out of wood, metal, plastic and glass. Thus, to classify something as a chair, the brain must take the image of the object projected onto the retina and convert it into an internal representation that remains invariant under all these conditions [1]. A lot of effort in psychology, computational neuroscience and computer vision has gone into understanding how the brain constructs these invariant representations [2, 3].

One hypothesis is that the brain learns these invariant representations from the statistics of natural images [2, 4, 5]. But until recently, it has proved challenging to construct scalable statistical inference models that learn directly from natural images and match human performance. A breakthrough has come in recent years from the field of artificial intelligence. Deep Convolutional Neural Networks (CNNs) are statistical inference models that are able to match, and in some cases exceed, human performance on some image categorisation tasks [6]. Like humans, these models show impressive generalisation to new images and to different translations, scales and viewpoints [7]. And like humans, this capacity to generalise seems to stem from the ability of Deep Networks to learn invariant internal representations [8]. It is also claimed that the learned representations in humans and networks are similar [7, 9, 10]. These results raise the exciting possibility that Deep Networks may finally provide a good model of human object recognition [11–14] and provide important insights into visual information processing in the primate brain [15–19].

Many reasons could be, and are, given for why CNNs have succeeded where previous models have failed [6, 20]. For example, it is often argued that CNNs excel in image classification because they incorporate a number of key insights from biological vision, including the hierarchical organization of the convolutional and pooling layers [21]. In addition, both systems are thought to implement optimisation frameworks, generating predictions by performing statistical inferences [18, 22]. Indeed, evidence suggests that humans perform some form of statistical optimisation for many cognitive tasks including language learning [23], spatial cognition [24], motor learning [25] and object perception [4]. Due to this architectural and computational overlap between the two systems it might seem reasonable to hypothesise that humans and CNNs end up with similar internal representations.

However, the parsimony and promise of this hypothesis is somewhat dampened by recent studies that have shown striking differences between CNNs and human vision. For example, CNNs are susceptible to small perturbations of images that are nearly invisible to the human eye [26–28]. They often classify images of objects based on statistical regularities in the background [29], or even based on single diagnostic pixels present within images [30]. That is,

CNNs are prone to overfitting, often relying on predictive features that are idiosyncratic to the training set [31].

To what extent do these findings reflect fundamental differences between CNNs and human vision? On the one hand, such differences could be simply down to differences in the learning environments of humans and artificial neural networks. Evidence supporting this hypothesis comes from studies that have compared features used by CNNs and humans to classify objects. Psychological experiments have repeatedly shown that humans rely primarily on global features, such as shape, for naming and recognising objects [32–35]. By contrast, a number of studies have demonstrated that CNNs trained on standard datsets, such as `ImageNet`, rely on local textures [36–38]. But these studies have also shown that, when CNNs are trained on datsets with the right biases, their behaviour can be brought a lot closer to human behaviour [37–40]. For example, Geirhos et al. [37] showed that CNNs trained on a modified version of `ImageNet` learn to show a shape-bias. Based on these results, Geirhos et al. conclude that "texture bias in current CNNs is not by design but induced by ImageNet training data". Similarly, Hermann et al. [38] showed that CNNs can learn to classify images based on global shape when they are trained with naturalistic data augmentations, leading them to conclude that "apparent differences in the way humans and ImageNet-trained CNNs process images may arise not primarily from differences in their internal workings, but from differences in the data that they see."

On the other hand, behavioural differences between humans and CNNs may arise out of more fundamental differences in resource constraints and mechanisms, rather than just differences in their training sets. While a CNN trained on a particular environment is able to mimic some aspects of shape-bias, the origin of shape-bias may be very different in the two systems. One way to distinguish between these two hypotheses is to check how the bias (of a network or human) is affected by moving to a new environment. If the origin of the bias is purely environmental, then a shift in the environment should also lead to a shift in the bias—that is, the system should start selecting features based on the statistical properties of the new environment. If, on the other hand, the bias is a reflection of a mechanistic principle or a resource constraint, it will be much more immune to a change in the statistical properties of the environment.

In this study, we explored this question by training models and humans to classify a set of novel objects. Each object contained multiple diagnostic features, all of which were clearly visible and could be used to perform the task. We manipulated the statistical bias for selecting these features, by manipulating the extent to which each feature type predicted the category labels. We wanted to explore the extent to which human adults and pre-trained CNNs were *adaptable* to the biases present within this task environment. At one extreme, people (and CNNs) could be completely adaptable, and select features solely based on the statistical properties of the new environment. At the other extreme, they could be completely inflexible and continue selecting features based on their prior biases. To gain a deeper insight into the role that prior biases play in learning new information, we compared the performance of both humans and CNNs to a statistical inference model that had no biases and learned to infer the category of a stimulus based on the sequence of samples observed in the task.

In a sequence of experiments that tested a range of different feature types and model settings, we observed that (i) the behaviour of human participants was in sharp contrast with the statistical inference model, with participants continuing to rely on global features, such as shape, and ignoring local features even when these features were better at predicting the target categories, (ii) when multiple global features were concurrently present (e.g. overall shape as well as colour), some participants chose to rely on one feature while others chose to rely on the second feature, but participants generally did not learn both features simultaneously, (iii) the behaviour of CNNs also contrasted with the statistical inference model, with the CNNs also

preferring to rely on one feature, (iv) however, unlike human participants, CNNs frequently relied on diagnostic local features and, crucially, this dependence on local features increased when the features were made more predictive, (v) CNNs were highly adaptable in the feature they used for learning—even when they were trained to have a shape-bias, this bias was lost as soon as they were trained on a new dataset with a different bias.

In two follow-up studies, we investigated whether human participants can overcome their bias for global features by (a) learning in an environment where there is no concurrent shape at all, or (b) being told what type of local feature to look for. In both cases, we observed that participants still failed to learn these tasks based on local features. Thus, the reason why participants ignore some clearly visible features is not simply due to the competition from shape, or to the difficulty in discovering these types of features. Rather, participants seem to struggle with the computational demands of learning the task based on certain features.

These results highlight important differences in how human participants and CNNs learn to extract features from objects and the role that existing biases play in adapting to novel learning environments. In general, CNNs are highly adaptable in learning new information, with the statistical structure of their learning environment driving their learning. While performing statistical learning is also clearly important for humans, their behaviour is much more strongly constrained by prior biases. Models of visual object recognition need to explain how such strong biases can be acquired and how they constrain learning in order to adequately capture human object recognition. The training and test sets developed in this study can be used to constrain and falsify models towards this end.

## Results

### Behavioural tasks and simulations

The behavioural tasks mimicked the process of learning object categorisation through supervised learning. In each experiment, participants were trained to perform a 5-way classification task, where they had to categorise artificially generated images into one of five categories. Each image consisted of coloured patches that were organised into segments. These segments were, in turn, organised so that they appeared to form a solid structure. Within each figure, the relative location, size and colour of patches as well as segments was perturbed (within some bounds) from image to image, making each stimulus unique and avoiding any unintentional diagnostic features, such as local features where segments intersect. In order to successfully perform the task, the participants and CNNs had to generalise over all these variables and discover the invariant shape or non-shape feature. See Fig 1 for some example images.

For each experiment, we constructed a dataset of images where one or more generative factors—*features*—predicted the category labels. In Experiments 1 to 4, images were drawn from datasets with two predictive features. One of these features was shape (the global configuration of segments) while the other feature was different in each experiment. In Experiment 1, the second feature was the location of a single patch in the image—that is, all images of a category contained a patch of a category-specific colour at a particular location (and none of the images from other categories contained a patch at this location). In Experiment 2, this feature was the colour of one of the segments—that is, all images assigned to a category contained a segment of a particular colour (and none of the images from other categories contained a segment of this colour). In Experiment 3, the second feature was the average size of patches—all patches in an image had similar sizes and the average size was diagnostic of the category. In Experiment 4, this feature was the colour of patches—all patches in an image had the same colour and images of different categories had different colours. In Experiment 5 and 6, all images had

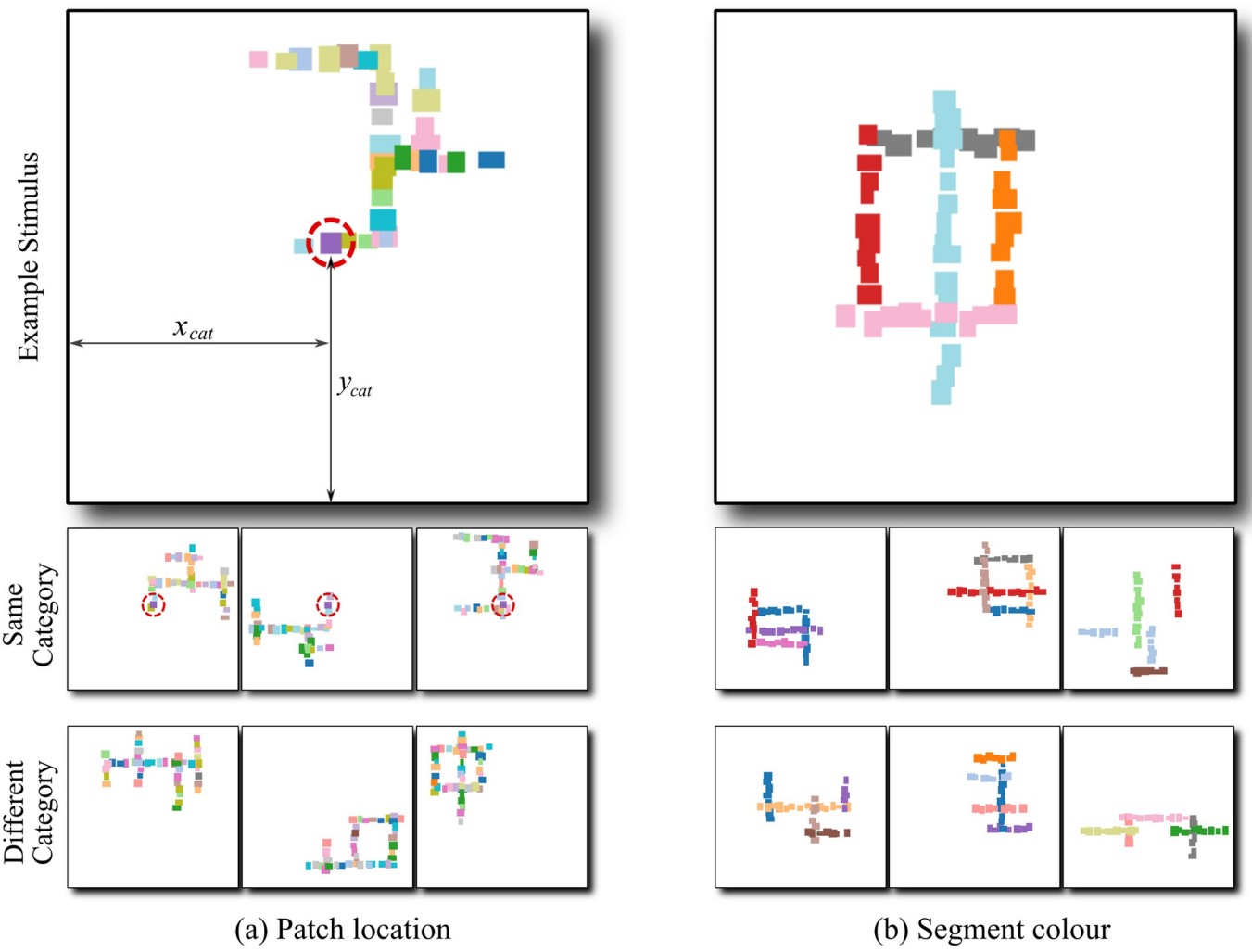

**Fig 1. Example training images from Experiments 1 and 2.** (a) Two features predict stimulus category: global shape and location ($x_{cat}$, $y_{cat}$) of one of the patches. For illustration, the predictive patch is circled. Stimuli in the same category (middle row, reduced size) have a patch with the same colour at the same location, while none of the stimuli in any other category (bottom row) have a patch at this location. (b) Global shape and colour of one of the segments predict stimulus category. Only stimuli in the same category (middle row) but not in any other category (bottom row) have a segment of this colour (red). The right-most stimulus in the middle row shows an example of a training image containing a non-shape feature (red segment) but no shape feature. For further illustration of stimuli used in these and other experiments, see S9–S13 Figs and S1 and S2 Movies.

only one predictive feature. This was either patch location, segment colour, patch size or overall colour; but none of the categories had a predictive shape.

Table 1 summarises the different combinations of features that were examined in the behavioural tasks in this study. Examples for all Experiments are shown in S9–S13 Figs and two movies illustrating the predictive features in Experiments 1 and 2 can be seen in S1 and S2 Movies.

In Experiments 1 to 4, training blocks were interleaved with test blocks which presented novel images that had not been seen during training. Each test block contained four types of test trials—`Both`, `Conflict`, `Shape` and `Non-shape`—that were designed to reveal the feature(s) used by the participant to categorise images. Trials in the `Both` condition contained the same combination of features that predicted an image's category during training. `Conflict` trials contained images with shape feature from one category and the second feature was swapped from another category. `Shape` trials contained images with only the shape

**Table 1. Feature combinations examined in different experiments.**

| Experiment | Features | | | | | % Shape |
|---|---|---|---|---|---|---|
| | Global Shape | Patch Location | Segment Colour | Average Size | Global Colour | |
| Exp 1a | ▓ | ▓ | | | | 100% |
| Exp 1b | ▓ | ▓ | | | | 80% |
| Exp 2a | ▓ | | ▓ | | | 100% |
| Exp 2b | ▓ | | ▓ | | | 80% |
| Exp 3a | ▓ | | | ▓ | | 100% |
| Exp 3b | ▓ | | | ▓ | | 80% |
| Exp 4a | ▓ | | | | ▓ | 100% |
| Exp 4b | ▓ | | | | ▓ | 80% |
| Exp 5 | | ▓ | ▓ | ▓ | ▓ | 0% |
| Exp 6 | | ▓ | ▓ | ▓ | | 0% |

Rows correspond to experiments and columns correspond to features. A shaded cell indicates that the feature in that column was used in the experiment in that row. The last column shows the proportion of training trials that contain a diagnostic shape. In Experiments 1–4 each participant saw stimuli that consisted of the combination of features shown in that row. Experiments 5 and 6 were between-subject designs so that participants were allocated to four (Experiment 5) or three (Experiment 6) groups and each participant saw stimuli with only one non-shape diagnostic feature.

feature and a non-predictive value of the second feature. Finally, the `Non-Shape` trials contained images where the five segments were placed at random locations on the canvas, giving the stimulus no coherent shape, but each image contained the second predictive feature (see S9–S13 Figs for some examples). An illustration of the four test conditions is shown in Fig 2. We measured accuracy in the `Both`, `Conflict` and `Shape` test trials based on the category predicted by the shape feature and accuracy in the `Non-shape` trials based on the category predicted by the non-shape feature.

We can infer the features that a participant uses by looking at the pattern of performance across the test conditions. There are four possible patterns. If a participant relies on shape, they should perform well in trials where shape predicts the image category. Thus their pattern of performance should be high, high, high, and low in the `Both`, `Conflict`, `Shape`, and `Non-shape` conditions, respectively. In contrast, if the participant relies on the non-shape feature, this pattern should be high, low, low, high. If a participant uses both (shape and non-shape) features, the pattern should be high, medium, high, high, where a "medium" performance in the `Conflict` condition is indicative of the fact that the two cues (features) learnt by the participant will compete with each other in these trials. Finally, if a participant does not learn either feature, their performance should be low in all four conditions. For a similar methodological approach for determining features used to categorise novel stimuli see [41].

In each experiment, we compared the behaviour of participants with two statistical inference models: an ideal inference model and a CNN. The ideal inference model computes what should a participant do if they had no prior biases and wanted to be statistically as efficient as possible, using all the information available during training trials. This model uses a sequential Bayesian updating procedure to compute the probability distribution over category labels given the training data and a test image. Similarly, the CNN computes the most-likely category-label for an image by learning a mapping between images in the training set and their category labels. Thus, it makes an approximate statistical inference by approximating a regressive model [42], (p85–89), but additionally has constraints built in through the choice of its architectural properties, such as performing convolutions and pooling. Both models are described in Materials and Methods below.

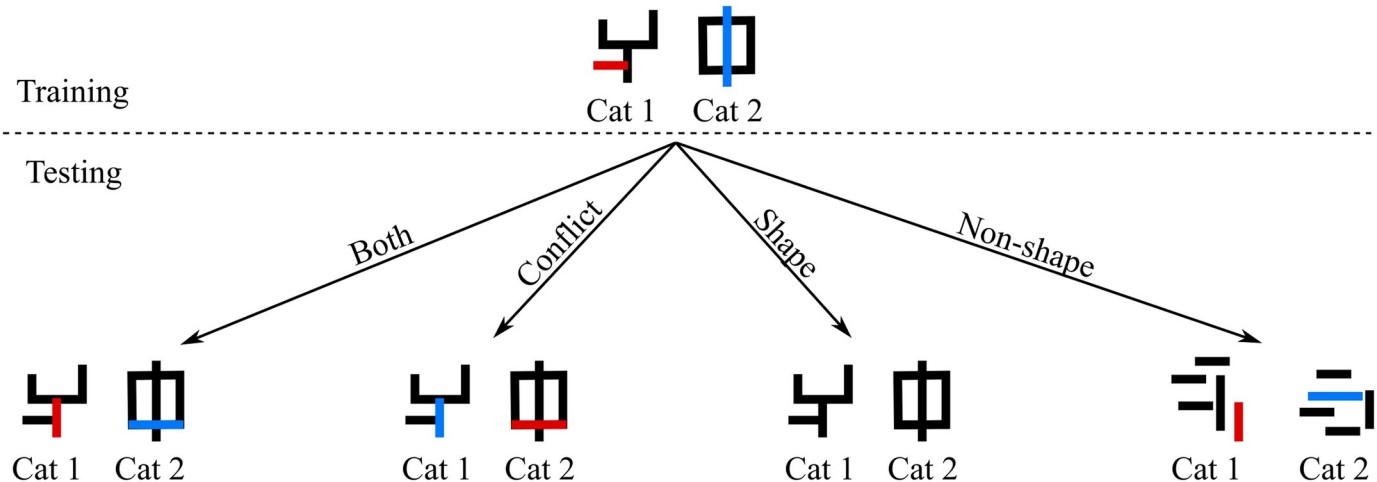

**Fig 2. An illustration of the four types of test conditions.** Each category has two diagnostic features: here, the overall shape and the colour of one of the segments. In the training images, features are mapped to categories using the following mapping: {(Shape A, Red) → Category 1; (Shape B, Blue) → Category 2}, where Shape A and Shape B are the shapes on the left and right, respectively. In the `Both` test condition, both types of features (shape and colour) have the same mapping as training. In the `Conflict` condition the mapping of the non-shape feature is swapped—i.e., the new mapping is {(Shape A, Blue) → Category 1; (Shape B, Red) → Category 2}. In the `Shape` condition, images have only one diagnostic feature—the overall shape—which has the same mapping as training: {Shape A → Category 1; Shape B → Category 2}. In the `Non-shape` condition, images have no coherent shape, but contain the same diagnostic colours as the training images: {Red → Category 1; Blue → Category 2}.

## Both features equally predictive

Fig 3A shows the pattern of performance in the final test block in Experiments 1a, 2a, 3a, and 4a. In these tasks, both shape and non-shape features perfectly and independently predict the category label during training. Thus the learner could use either (or both) features to learn an image's category. The top row shows the pattern of performance for the ideal inference model. In all four tasks, this model predicts that the probability of choosing the correct category is high in the `Both`, `Shape` and `Non-shape` conditions and significantly lower in the `Conflict` condition. This indicates that there is enough information in the training trials for all four experiments to predict the category label based on either the shape or the non-shape feature.

Note that the ideal-inference model predicts that the accuracy for the `Conflict` condition is different for Experiment 1a from the other experiments. The shape and non-shape cues are equally competitive in Experiments 2a, 3a and 4a. Consequently the probability of choosing the correct (shape-based) category is around 0.50 in the `Conflict` condition in these experiments. However, the results in Fig 3A show that, in Experiment 1a, the non-shape cue dominates the shape cue in the `Conflict` condition. This is because an image with a diagnostic patch at one of the diagnostic locations contains two types of information: (i) a diagnostic colour at one of the diagnostic locations, and (ii) white (background) patches at all the other diagnostic locations. These two signals together dominate the shape signal in `Conflict` trials in Experiment 1.

The middle row shows the pattern of performance for the CNN model. In all four tasks, the network showed high accuracy in the `Both` condition—showing an ability to generalise to novel (test) stimuli, as long as both shape and non-shape features were preserved in the stimuli. It showed a low accuracy in the `Conflict` condition, but high accuracy in the `Non-shape` condition. Its performance in the `Shape` condition was above chance in Experiments 1a, 2a and 3a and at chance in Experiment 4a. The above-chance performance in the `Shape`

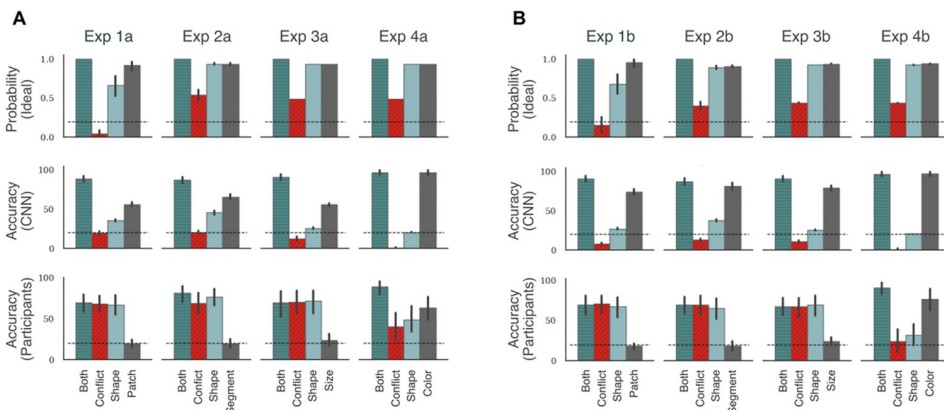

**Fig 3. Results in Experiments 1–4.** Each column corresponds to an experiment and each row corresponds to the type of learner (ideal inference model, CNN or human participants). The top row shows the posterior probability of choosing the labelled class for a test trial given the training data. The bottom two rows show categorisation accuracy for this labelled class. Panel A shows results for experiments where both features are equally predictive (1a, 2a, 3a and 4a), while Panel B shows results for experiments where the non-shape feature is more predictive (1b, 2b, 3b and 4b). Each plot shows four bars that correspond to the four types of test trials. `Patch`, `Segment`, `Size` and `Colour` refer to the `Non-shape` test trials in Experiments 1, 2, 3 and 4, respectively. Error bars show 95% confidence and dashed black line shows chance performance. In any plot, a large difference between the `Both` and `Conflict` conditions shows that participants rely on the non-shape cue to classify stimuli. Both models show this pattern while humans show no significant difference.

condition implies that this network is able to pick up on shape cues. However, its performance is significantly lower in the `Shape` condition compared to the `Non-shape` condition. When these two cues competed with each other, in the `Conflict` condition, the network favoured the non-shape cue and the accuracy was at or below chance. These results indicate that the CNN learns to categorise using a combination of shape and non-shape features.

It is also worth noting that, unlike the ideal inference model, the CNN showed a bias towards relying on non-shape features in all experiments, even though it would be ideal (from an information-theoretic perspective) to learn both features in parallel. A similar result was observed by Hermann et al. [40], who found that when multiple features predict the category, CNNs preferentially represent one of them and suppress the other.

The bottom row shows the average accuracy in the four experiments for human participants (N = 25 in each task). Like the ideal inference model and the neural network model, participants showed high accuracy in the `Both` condition (mean accuracy was between 70% (in Experiment 1a) and 89% (in Experiment 4a). This indicates an ability to generalise to novel (test) stimuli as long as shape and non-shape features were preserved. However, their pattern of performance across the other three conditions were in sharp contrast to the two models. In Experiments 1a, 2a, and 3a, participants showed a high-high-high-low pattern in the `Both-Conflict-Shape-Non-shape` conditions, indicating that they strongly preferred the shape cue over the non-shape cue. In fact, performance in the `Non-shape` trials was at chance in all three tasks with mean accuracy ranging from 20% to 24%. Single sample t-tests confirmed that performance was statistically at chance in all three tasks (largest $t(24) = 0.99$, $p > .05$). Thus, unlike the ideal inference model, which learnt both predictive cues, participants chose one of these cues. And unlike the neural network model, which favoured the non-shape cue, participants preferred to rely on shape.

The behaviour of participants was different in Experiment 4a, where the non-shape cue was the colour of the entire figure. Performance was again high in the `Both` condition, but

significantly lower in the `Conflict`, `Shape` and `Non-shape` conditions ($F(3, 72) = 8.18$, $p < .01, \eta_p^2 = .25$). So, on average, participants seemed to be using both shape and non-shape (colour) cues to make their decisions, but neither feature was strongly preferred over the other. This behaviour seemed to be qualitatively similar to the ideal inference model, which learnt to use both predictive cues simultaneously. However, examining each participant separately, we found that participants could be grouped into two types, those that primarily relied on shape (N = 12) and those that relied on colour (N = 13). Participants were categorised as relying on colour if performance in the `Non-shape` condition was above performance in the `Shape` condition. S4 Fig shows the average pattern of performance for each of these groups. The first group shows a high-low-low-high pattern, indicating that they were predominantly using the colour cue to classify test images. The second group shows a high-high-high-low pattern, indicating that they were predominantly using the shape cue. Mixing these two groups of participants results in the high-medium-medium-medium pattern shown in Fig 3A.

## One feature more predictive than the other

Our next step was to check what happens when one of the features predicts the category *better* than the other. If the nature of shape-bias is similar in humans and CNNs, we expect both systems will adapt in a similar way to a new statistical environment, which favours a non-shape feature. In Experiments 1b, 2b, 3b, and 4b the shape feature predicted the category label in only 80% of the training trials. The remaining 20% images contained horizontal and vertical segments placed at random locations on the canvas so that these images contained no coherent shape. The second feature (patch location, segment colour, patch size or overall colour) predicted the category label in 100% of training trials. See Fig 1 and S13 Fig for some examples of training images that do not contain a shape feature but contain a non-shape feature. Fig 3B shows the performance for the two models as well as human participants (N = 25 in each task). The ideal inference model (top row) showed a very similar performance, again predicting that a participant should learn both features simultaneously. Its accuracy on non-shape feature was slightly better. This is a consequence of larger number of samples containing non-shape cues. In contrast, the performance for the CNN model was significantly different. In all experiments, the model now showed a high–low–low–high pattern, with performance in the `Shape` condition close to chance in most experiments. Thus, the CNN model started relying almost exclusively on the (more predictive) non-shape feature.

In contrast to both models, participants continued showing a high–high–high–low pattern in Experiments 1b, 2b, and 3b, indicating a clear preference for relying on shape. It should be noted that this happens even though shape is *not* the most predictive feature. In fact, performance in the `Non-shape` condition was at chance (mean accuracy ranged from 18% to 24%, largest $t(24) = 1.74$, $p > .05$ when compared to chance level), showing that participants completely ignored the most predictive feature.

The behaviour of participants was again different in the experiment using colour of entire figure as the non-shape cue (Experiment 4b). Average accuracy across participants was high in the `Both` condition, but significantly lower in the `Conflict`, `Shape`, and `Non-shape` conditions ($F(3, 72) = 22.68, p < .01, \eta_p^2 = .49$). Like Experiment 4a, examining each participant separately in Experiment 4b showed that participants could be divided into two groups— those that learnt to rely on shape and those that learnt to rely on colour. However, the ratio of participants in these groups changed. While 12 participants (out of 25) relied on shape in Experiment 4a, 7 participants (out of 25) relied on it in Experiment 4b (see S5 Fig).

## Effect of previous training on CNN behaviour

In the above experiments, we observed that the participants systematically deviated from the two statistical inference models. This contrast was particularly noteworthy in Experiments 1b-4b. Here, the non-shape feature was more predictive than shape but participants still focused on global features like shape. In contrast, the CNN preferred to rely on the more predictive (non-shape) feature. So we wanted to explore whether CNNs can be made to behave like humans through training. A recent set of studies have suggested that CNNs indeed start showing a shape-bias if they are pre-trained on a dataset that contains such a bias [37, 38]. However, after the network had been pre-trained on the first set with a shape-bias, these studies did not systematically manipulate how well each feature predicted category membership in the new set of images. This is a crucial manipulation in the above studies that allowed us to more directly assess the feature biases of CNNs, and our results suggest that the CNN learns to rely on the most diagnostic feature in this new set.

To test the effect of pre-training, we used the same CNN as above—`ResNet50`—but this time pre-trained on the Style-transfer ImageNet database created by [37] to encourage a shape-bias. We then trained this network on our task under two settings: (i) the same setting as above, where we retuned the weights of the network at a reduced learning rate, and (ii) an extreme condition where we froze the weights of all convolution layers (that is 49 out of 50 layers) limiting learning to just the top (linear) layer.

The results under these two settings are summarised in Fig 4. In line with previous results [37, 38], we observed that this network had a larger shape-bias—for example, it predicts the target category better in the `Shape` condition than the network pre-trained on ImageNet (compare with the middle row in Fig 3). In some cases, this makes the network behave more like the ideal inference model, where it is able to predict the category based on either shape or non-shape features. But this pattern is still in contrast with participants who were at chance when predicting based on non-shape features in Experiments 1–3. Crucially, when the non-shape feature is made more predictive, the network shows a bias towards this feature, showing the same high-low-low-high pattern observed above (Fig 4, top right). Even under the extreme condition, where we froze the weights of all except the final layer, the network preferred the non-shape feature as long as this feature was more predictive (Fig 4, bottom right). That is, CNNs do not learn to preferentially rely on shape when learning new categories even when pre-trained to have a shape bias on other categories.

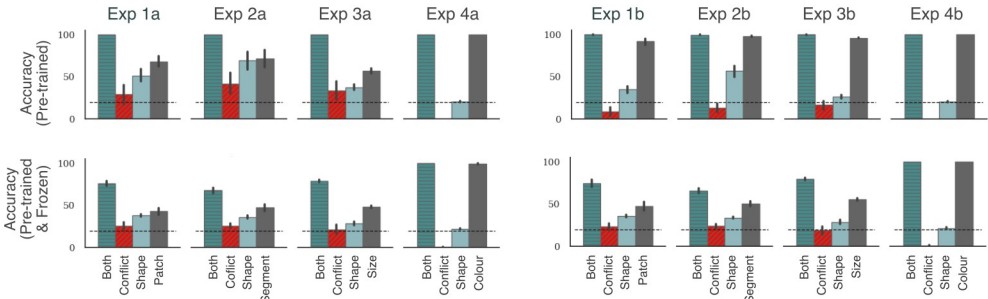

**Fig 4. Results for pre-training on a dataset with a shape-bias.** The first row shows results when the CNN was pre-trained on the Style-transfer ImageNet [37] and allowed to learn throughout the network. The second row shows results of the same network when weights for all convolution layers are frozen. First column shows results when both features are equally likely (Experiments 1a, 2a, 3a and 4a) while the second column shows results when the non-shape cue is more predictive (Experiments 1b, 2b, 3b and 4b). In all panels, we again observed a large difference between the `Both` and `Conflict` conditions, indicating that despite pre-training, models relied heavily on the non-shape cue to classify stimuli.

## Dynamics of learning

We probed the learning strategy used by models and participants by examining performance at regular intervals during training. If a participant (or model) learns multiple features in parallel, they should show an above-chance performance on both the `Shape` and `Non-shape` test trials at the probed interval. If they focus on a single feature, their performance on that feature should be above-chance and match the performance on the `Both` trials. If they switch between different features over time, their relative performance on `Shape` and `Non-shape` trials should also switch over time.

Fig 5A shows the performance under the four test conditions over time for Experiments 1b, 2b, 3b and 4b (results for Experiments 1a, 2a, 3a and 4a show a similar pattern and are shown in S6 Fig). The ideal inference model shows an above-chance performance on the `Shape` as well as `Non-shape` trials throughout learning. This confirms the expectation that the ideal inference model should keep track of both features in parallel. However, this is neither what the CNN nor what human participants do. The CNN shows a bias towards learning the most predictive (non-shape) feature from the outset, with performance on the `Non-shape` trials closely following performance on the `Both` trials. Human participants showed the opposite bias, with performance on the `Shape` trials closely following performance on the `Both` trials. We did not observe any case where the relative performance on the `Shape` and `Non-shape` trials switched over time. This suggests that participants did not systematically explore different features and choose one—rather they continued learning a feature as long as it yielded enough reward. Even in Experiment 4b, where some participants used the colour cue while others used the shape cue, no participant in either group showed any evidence for switching form one feature to the other.

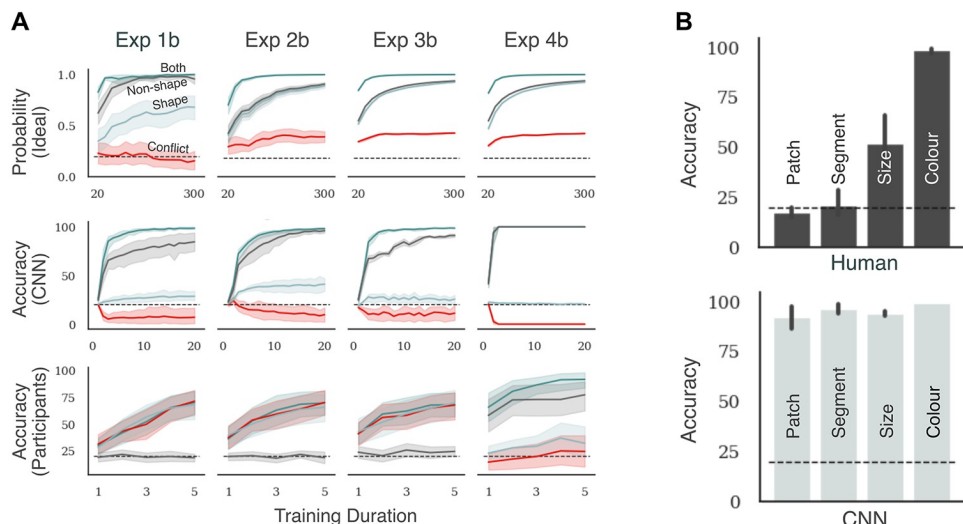

**Fig 5. Each plot in Panel A shows how accuracy on the four types of test trials changes with experience.** The top, middle and bottom row correspond to ideal inference model, CNN and human participants respectively. Columns correspond to different experiments. The scale on the x-axis represents the number of training trials in the top row, the number of training epochs in the middle row and the index of the test block in the bottom row. The two plots in Panel B show accuracy in test blocks for humans and CNN, respectively, when they are trained on images that lack any coherent shape. Each bar corresponds to the type of non-shape feature used in training.

## Learning in the absence of shape

The above experiments always pit a highly predictive feature against shape. We wanted to know whether participants struggle to learn the predictive local feature even when a diagnostic shape was absent. If participants only fail to learn this feature when a diagnostic shape is present, it indicates a difference in the bias between participants and CNNs (humans prefer global shape, while CNNs prefer more local features). On the other hand, if participants struggle to learn this feature even when it is clearly visible and a diagnostic shape is absent, it indicates a more fundamental limitation in human (but not CNN) capacity to extract these features. To test this, we designed a behavioural task (Experiment 5) where a shape feature was absent from the training set. Like the above experiments, each training stimulus still contained a set of patches and segments, but the segments were not consistently organised in a spatial structure (see S13 Fig for examples of this stimuli). Instead, every training trial contained a non-shape predictive feature. We used the same features as above—patch location, segment colour, patch size or overall colour. Participants were divided into four groups based on the type of predictive feature they were shown in the training trials. The test block consisted of novel images (that were not seen in training) but had the same diagnostic feature as training (equivalent to the `Non-shape` condition in the above experiments).

The average accuracy in test trials for each type of diagnostic feature is shown in Fig 5B. There was a large difference in performance depending on the type of diagnostic feature. When the colour of the entire figure predicted the category, accuracy on test trials was high ($M = 98.67\%$). The responses collected for training trials indicated that participants learned this feature quickly (performance reached 94.40% after 100 training trials). Accuracy in the test block was lower (though still significantly above chance) when the size of patches predicted the category ($M = 52.40\%$) and participants learned this feature at a slower rate. In contrast to these two conditions, participants were unable to learn the other two diagnostic local features. Performance was at chance in test trials both when the colour of a segment predicted the category ($M = 21.47\%$) and when the location and colour of a single patch predicted the category ($M = 17.47\%$). Thus participants seemed sensitive to the computational complexity of the diagnostic feature. They extracted simple features like the colour of the entire figure or the size of patches, but did not extract more complex features like colour of single segment or patch. Fig 5B also shows the performance of the CNN on this task. In contrast to human participants, the network learnt all four types of non-shape stimuli and showed high accuracy on test trials in all four conditions.

## Identifying versus learning features

In order to discover the correct diagnostic features in the experiments above, a participant must perform two distinct operations: they must identify a diagnostic feature (from a list of all possible features) and match the correct value of this feature to each category. For example, in Experiment 2, the participant must first realise that the diagnostic feature is the colour of each segment. That is, they must find this feature in the space of all possible features (shape, number of patches, location, size, etc.). Secondly, they must map the stimulus on a given trial to the correct category, extracting the colour of all five segments, working out which segment is diagnostic and what the mapping is between the diagnostic colour and category. The second operation—mapping a diagnostic value to a category—is a computationally demanding task as it requires the participant to remember several pieces of information, comparing the features observed in a given stimulus with the features and outcomes of past stimuli. One reason why participants might fail when the CNN succeeds is that humans and CNNs have very different computational resources available to them. For example, while humans are limited by the

capacity of their working memories (the number of features they can process at the same time), CNNs have no such limitations. If this was the case—i.e., if participants were failing because of their limited cognitive resources and not because they were unable to identify the correct feature—we hypothesised that helping the participants identify the diagnostic feature will not improve their performance on these tasks.

We checked this hypothesis in Experiment 6 that repeated the design of Experiment 5, where participants saw stimuli that had only the non-shape diagnostic feature and no coherent shape. Instead of letting participants figure out which feature was diagnostic, we informed them of the diaganostic feature in each task and showed them two examples of stimuli with the diagnostic feature (see Materials and methods for details). Additionally, we increased the duration of each stimulus from 1s to 3s to ensure that participants do not underperform because of the time constraint. Finally, we gave participants an added incentive to learn the task, increasing the possible bonus reward based on their performance in the test block. Participants then completed 6 training blocks (50 trials each) where they saw random samples of stimuli from each category. We already know that participants can solve the task when the diagnostic feature was the colour of the entire figure (see Fig 5B above). Therefore, we tested three groups of participants, where each group was trained on stimuli with one of the other three non-shape features—patch location, segment colour or average size—being diagnostic of the category.

The results of Experiment 6 are shown in Fig 6. Like Experiment 5, mean performance across participants was above chance in the Size condition but at chance in the Patch and Segment conditions. The overall pattern of results for the three conditions was statistically indistinguishable from the results of Experiment 5. In other words, even when participants were told the diagnostic features and given additional time and incentive to learn the task, they struggled to classify stimuli based on patch location or segment colour. These results confirm the hypothesis that the difficulty of these tasks for human participants is not limited to

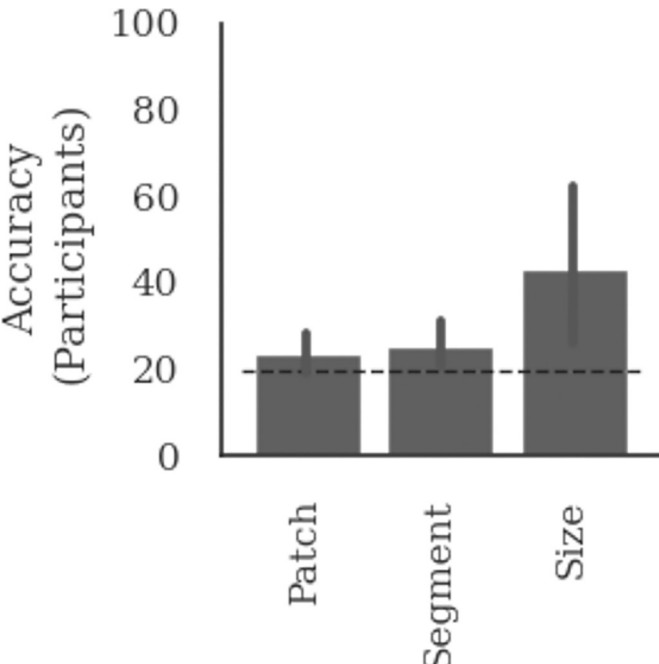

**Fig 6. Results of telling participants the diagnostic cue.** Each bar shows mean accuracy across 10 participants in the test block. Participants were divided into three groups based on the diagnostic cue—patch location, segment colour, or average size—used to train the participants.

identifying the diagnostic features. Instead, the cognitive resources required to extract the diagnostic feature value and mapping it to the correct category may play a critical role in how humans select features for object classification.

## Discussion

In a series of experiments we repeatedly observed that participants learned to classify a set of novel objects on the basis of global features such as overall shape and colour even when local non-shape features were more predictive of category (Fig 3B). This behaviour is in keeping with psychological studies which show that humans prefer to categorise objects based on shape [32–34, 43] but, additionally, shows that this shape-bias is retained in novel learning environments where the statistics favoured learning based on a different feature. This observation is consistent with category-learning studies which show that participants overlook salient cues when multiple cues can be used to solve the task [44] and especially in high-dimensional classification tasks [45].

We found that one cannot explain human behaviour using a simple statistical model that infers the category of a test stimulus based solely on the evidence observed in the training trials and no prior biases. We also found that human behaviour was inconsistent with the behaviour of CNNs as the the predictive value of features play a key role in how CNNs learns to classify novel objects. Unlike human participants, previous biases of the network (either learnt through training or built-in through architectural constraints) were not sufficient to overcome this reliance on predictive features. If humans indeed learn in novel environments through a process of statistical learning, these results motivate an exploration of why humans do not quickly adapt to the novel environment in the same way the statistical models presented in this study do. Note that this may be a challenging problem to solve for CNNs and statistical inference models as in Experiment 5 and 6 participants struggle to learn some features even when there is no concurrent shape feature.

Our results were robust across a range of experimental and simulation conditions. Of course, this does not mean that we have controlled for all differences between human experiments and CNN simulations, but our study shows that our findings are robust across multiple CNN architectures, a range of hyper-parameters, different types of pre-training and different types of predictive features (patch location, segment colour, patch size). While we believe that there is unlikely to be a set of experiment conditions that will make humans behave like CNNs, we acknowledge that we do not control for all possible differences in the experimental setup.

Another difference between human participants and CNNs is that participants in our studies had a life-time of exposure to a natural world where shape may be the most diagnostic feature. Indeed, some longitudinal studies have shown that it is possible to create a shape-bias in very young children by intensively teaching them new categories but keeping the statistical properties of their linguistic environment [46]. Accordingly, it is possible that our participants had acquired a shape-bias early on in life [35, 47] that constrained how the new objects in our experiments were learned. But we observed that CNNs did not *retain* a shape-bias even when we induced a shape-bias in pre-training and when we froze the weights in an attempt to preserve the shape-bias when classifying our new objects. Instead they simply learned whatever features of new object categories were most diagnostic. In other words, even if one assumes that CNNs adequately capture why humans learn to classify objects based on their shape, they do not capture why humans continue to look for certain features (like shape) and are agnostic to other features when learning about new objects.

Note, we do not want to claim that humans could never learn to use features other than shape. In Experiment 4, many participants learn to rely on another global feature—the overall

colour of objects. And in some of our experiments (for example, where the size of the patches predicted category membership) it is possible that if participants were given a lot more training, some participants may switch to using the more predictive feature. If this were the case, the pattern would be that participants prefer relying on global features early in learning, then switch to more predictive features. The dynamics of the ideal inference model and the CNN (Fig 5A) show that neither of the models predict this behaviour.

It should also be noted that the behaviour of participants observed here highlights a more extreme form of shape-bias than has been reported before. In a typical shape-bias experiment, the term shape-bias indicates the inductive-bias to rely on shape in the presence of alternative features that are equally good at predicting the target category [34, 35]. In our experiments, we observed that participants relied on shape even in the presence of features that were *better* at predicting the target category. Furthermore, in two of our experiments (Experiments 5 and 6) there was no consistent shape at all that could be used to predict category membership. In these experiments, participants failed to pick some perfectly predictive statistical features (like location of patch or colour of segment) even in the absence of a diagnostic shape. This functional blindness towards certain features cannot necessarily be explained as a shape-bias as there is no competing shape feature to learn.

These findings are consistent with a recent study conducted by Shah et al. [48], who found that CNNs learn to classify images on the basis of simple diagnostic features and ignore more complex features. The focus of Shah et al. [48] was not on comparing CNNs to humans, but rather, showing how a simplicity bias limits generalisation in CNNs. Nevertheless, their study may shed light on another key difference between human and CNN vision, namely, humans are much better at generalising to out-of-distribution image datasets compared to CNNs, such as identifying degraded and distorted images [31, 49]. It may be that that the shape bias we observed in humans but lacking in CNNs plays a role in more robust human visual generalisation.

An important outstanding question is *why* participants in our study relied on global features such as shape or overall colour and struggled to learn salient features that were highly diagnostic. Some of the observations made in our experiments provide clues to the reasons underlying participant behaviour. In Experiment 6, we observed that even when the relevant features are pointed out, participants still could not learn to classify objects based on patch location and segment colour. This shows that the inability to learn these local features is not limited to the difficulty of discovering the type of feature, but may be due to the computational demand of learning how features map to categories. For example, consider the Segment condition in Experiment 6, where the colour of a segment predicted the object category. One strategy to learn this task is to simultaneously store colours of all five segments in memory during each trial and compare these colours across trials of the same category, eliminating colours that do not overlap. This type of strategy will have strained or exceeded the visual capacity of humans, leading them to ignore this predictive cue and focus on shape, even though it is less diagnostic.

Similarly, we also observed that participants frequently selected only one of several possible features available to learn an input-output mapping (e.g. in Experiment 4 participants chose to classify either based on colour or shape but almost never both, even though this was the optimal policy in the task). Learning multiple features may lead to better prediction in certain circumstances, however it also requires using more cognitive resources. The fact that participants generally rely on only one feature suggests that participants trade off their performance in the task with the mental effort [50, 51] required to learn how each feature maps to the object category.

By contrast, CNNs do not suffer from the same resource limitations as humans. A striking example of this is that CNNs not only succeed in learning to classify millions of images in `ImageNet` into 1000 categories, they can also learn to classify the same number of random patterns of TV static-like noise into 1000 different categories [52], something far beyond the capacity of humans [53]. This capacity was no doubt exploited by the CNNs in the current learning context. By contrast, our participants had to learn the object categories in the face of many well documented cognitive limitations of humans, such as limited capacity of visual short-term memory [54], visual crowding [55, 56] and selective attention [57, 58].

Whatever the origin of the shape-bias, the results here should give pause for thought to researchers interested in computational models of visual object recognition. These results show that humans are blind to a wide range of non-shape predictive features when classifying objects, and if models are going to be used as theories of human vision, they should be blind to these features as well. This may result not only in models that are more psychologically relevant, but also capture the robustness and generalisability of the human visual system that is lacking in current models [28, 31, 59].

## Materials and methods

### Ethics statement

All studies adhered to the University of Bristol ethics guidelines and obtained an ethics approval from the School of Psychological Science Research Ethics Committee (approval code 10350). For all behavioural experiments, we obtained formal written consent from participants to use their anonymised data for research.

### Experimental details

**Materials.** We constructed nine datasets of training and test images. There were 2000 training images and 500 test images in each dataset. Each image consisted of 30–55 coloured patches on a white background. The colours of patches were sampled from a palette of 20 distinct colours so that they were clearly discernible. These patches were organised into five segments. There were four short segments (consisting of 5–10 patches) and one long segment (consisting of 10–15 patches). Each segment was oriented either vertically or horizontally. Images were grouped into five target categories and each category was paired with a unique spatial configuration of segments. It is this spatial configuration of segments that we refer to as *shape*. These shapes were chosen such that the five shapes were clearly distinct from one another. A pilot experiment showed that most participants could learn to categorise based on the chosen shapes within 300 trials. All images in a category also contained a second diagnostic feature, which was the location and colour of a patch in Experiment 1, the colour of a segment in Experiment 2, the average size of patches in Experiment 3 and the colour of all the segments in Experiment 4.

Within each category, images were randomly generated and varied in the number, colour, location and size of patches. This variability ensured that (i) participants (human and CNN) had to generalise over images to learn the category mappings, and (ii) there were no incidental local features that could be used to predict the category. The exact number of patches in each segment was sampled from a uniform distribution; the size and location of each patch was jittered (around 30%); and the colour of each patch (Experiments 1 and 3) or each segment (Experiment 2) was randomly sampled from the set of (non-diagnostic) colours. In addition, each figure was translated to a random location on the canvas and could be presented in one of four different orientations (0, $\pi/2$, $\pi$ and $3\pi/4$ radians).

The original size of images was 600x600 pixels. This was reduced to 224x224 pixels for the simulations with CNNs. For the behavioural experiments, the stimuli size was scaled to 90% of the screen height (e.g. if the screen resolution was 1920x1080 the image size would have been 972x972). This ensured that participants could clearly discern the smallest feature in an image (a single patch) which we confirmed in a pilot study (see Procedure below).

**Participants.** Participants were recruited and reimbursed through Prolific. In Experiments 1–4, S1 and S2 there were N = 25 participants per experiment (total N = 250), and in Experiments 5, 6, S3 and S4 there were N = 10 participants per experimental condition (total N = 100). In Experiments 1–5 as well as S1-S3 participants received 4 GBP for participating in the experiment and could earn an additional 2 GBP depending on average accuracy in the test blocks. In Experiment 6 and S4 the incentive was increased to 5.30 GBP and a possible bonus of 3 GBP based on performance in the test block. Calculated as payment per hour, the average payout per participant in our experiments was 7.62 GBP per hour.

**Procedure.** All experiments consisted of blocks of training trials, where participants learned the categorisation task, followed by test trials, where their performance was observed. During training trials participants saw an image for a fixed duration and were asked to predict its category label (see Fig 7). In Experiments 1–5, this duration was 1000 ms, but we experimented with both longer durations (Experiments 6 and S4) and shorter durations (Experiments S1–S3, see below) and obtained a similar pattern of results. After each training trial, participants were told whether their choice was correct and received feedback on the correct label if their choice was incorrect. In Experiments 1 to 5, participants had to discover the predictive features themselves, while in Experiment 6, they were explicitly told what the predictive feature was at the beginning of the experiment. In this experiment, they were given textual instructions describing the target feature and shown exemplars where the target feature was highlighted. Participants saw 5 blocks of 60 training trials in Experiments 1–4 and 10 blocks of

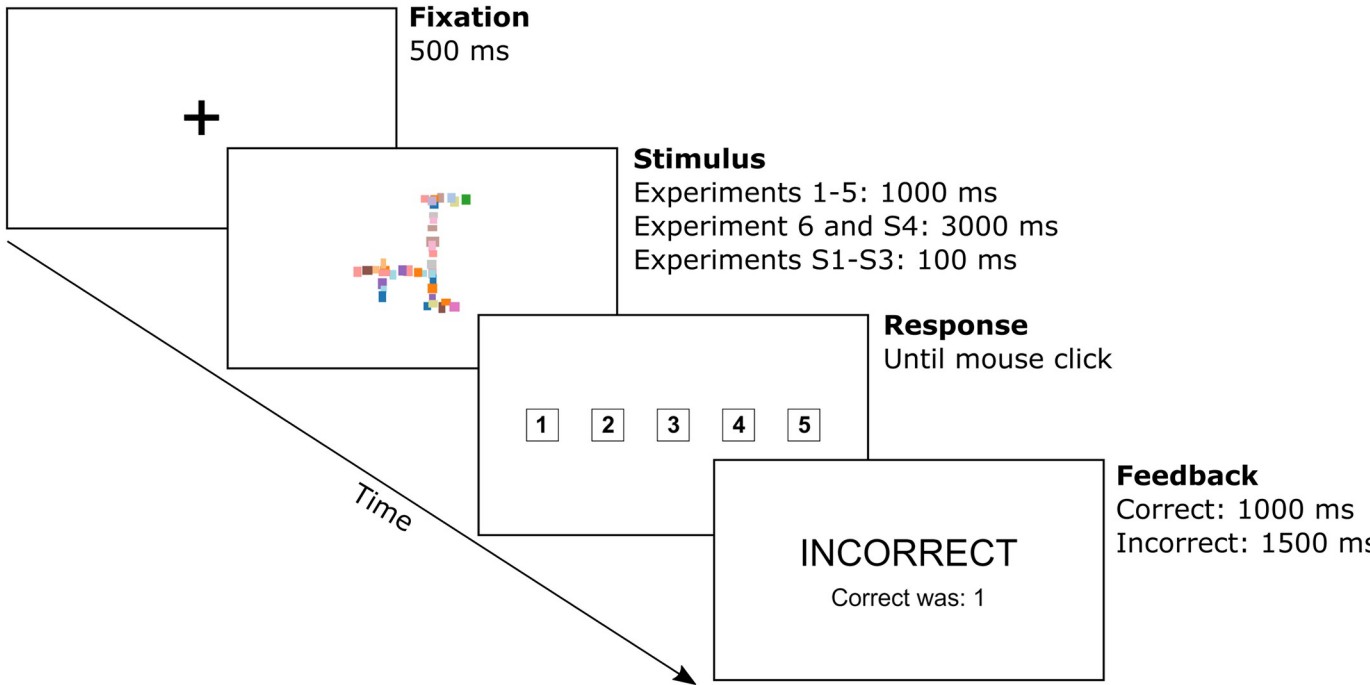

**Fig 7. Procedure for human experiments.** Time course for a single training trial in human experiments. The test trials followed an identical procedure, except participants were not given any feedback on their choices.

50 trials in Experiments 5 and 6. The number of training trials was chosen based on a pilot experiment and ensured that participants learnt the behavioural task. In Experiments 1 to 4, each training block was followed by a test block containing 40 trials (10 per condition). In Experiments 5 and 6, one test block was presented at the end of training consisting of 75 trials. Test trials followed the same procedure as training, except participants were not given any feedback. As we were interested in object recognition rather than visual problem solving, all trials (training as well as test) used a short presentation time of 1000ms. In a follow-up experiment (as well as Experiment 6), we also tried a longer presentation time of 3000ms and observed a similar pattern of results (see S8 Fig).

All experiments were designed in PsychoPy and carried out online on the Pavlovia platform. We ensured that participants could clearly see the location of each patch by conducting a pilot study. In this study, participants were shown an image from one of our datasets and asked to attend to a highlighted patch. After a blank screen they were shown a second image from the same dataset and asked to click on the patch which was in the same position as the highlighted patch in the first image. We found that the median location indicated by participants deviated from the center of the target patch by only a quarter of the width of a patch—meaning that participants were able to attend, keep in working memory and point out a specific patch location. This indicates that even the smallest of the local features used in this study was perceivable for human participants.

In order to ensure that our results are not affected by the presentation time or field of view, we conducted three control experiments. The results of our main experiments (see Fig 3) showed that be biggest contrasts between participants and humans were in Experiment 1 and 2, where the diagnostic non-shape feature was the location of a patch or the colour of a segment. Therefore, we conducted three control experiments, reproducing the setup of Experiments 1, 2 and 5 (Patch and Segment conditions). All details of these control experiments were the same as above, except (i) presentation time of stimulus was reduced to 100ms, (ii) the stimulus was re-scaled so that it was always within 10° visual angle, (iii) instead of testing participants in between every training block, we tested participants only at the end, and (iv) in order to ensure that participants are able to learn the task despite the shorter presentation time, we increased the number of training trials from 300 to 450. We re-scaled the stimulus by using the ScreenScale script https://pavlovia.org/Wake/screenscale, which has been shown to give good estimates of visual angle in online experiments [60]. Participants were asked to adjust the size of a displayed rectangle to the size of a credit card. To ensure that participants did this correctly, we asked participants to measure the size of a second rectangle and rejected all participants whose measurements did not match the correct size. To compute the visual angle, we asked participants to sit at an arm's length from the screen and asked them to measure the distance between their eyebrow and a fixation cross on the screen. Based on this measurement and how participants re-scaled the displayed rectangle, we re-scaled the stimulus so that the entire image subtended a visual angle of 10°. Participants were reminded to sit at an arm's length at the end of every training block. The results of these control experiments are shown in S7 Fig.

**Data analysis.** In all experiments chance performance was 20% since there is a 1 in 5 chance of randomly picking the correct category. Single sample t-tests were conducted in order to determine whether participants were above chance level performance. Repeated measures analyses of variance (ANOVA) were conducted when determining whether there was an effect of condition (Both, Conflict, Shape, Non-shape) on performance in an experiment. Follow-up comparisons were conducted with the Tukey HSD correction for multiple comparisons.

## Simulation details

**Neural network model.** During a supervised learning task (like the task outlined in this study), a neural network performs an approximate statistical inference by constructing an input-output mapping between a random vector $\mathbf{X}$ and a dependent variable $Y$. The training set consists of $N$ realisations of this random vector, $\{\mathbf{x_1} \ldots, \mathbf{x_N}\}$ and $N$ category labels $\{c_1 \ldots, c_n\}$. For a CNN, the vectors $\mathbf{x_i}$ can simply be an image (i.e. a vector of pixel values). That is, $\mathbf{X}$ lies in a high-dimensional image space. The neural network learns a non-linear parametric function $\hat{c}_i = F(\mathbf{x_i}, \mathbf{w})$ by finding the connection weights $\mathbf{w}$ which minimise the difference between the outputs produced by the network $\hat{c}_i$ and the given category labels, $c_i$. During a test trial, the network performs an approximate statistical inference by deducing the class of a test vector $\mathbf{x_{test}}$ by applying the learnt parametric function to this vector: $c = F(\mathbf{x_{test}}, \mathbf{w})$.

Since our task involved image classification, we evaluated three state-of-the-art deep convolutional neural networks, ResNet50 [61], VGG-16 [62] and AlexNet [63] which performs image classification on some image datasets to a near-human standard. We obtained the same pattern of results with all three architectures. Therefore, we focus on the results of ResNet50 in the main text and describe the results of the other two architectures in S1–S3 Figs. Since evolution and learning both play a role in how the human visual system classifies natural objects, we used a network that was pre-trained on naturalistic images (`ImageNet`) rather than trained from scratch. However, we observed the same pattern of results for a network that was trained from scratch. In each experiment, this pre-trained network was fine-tuned to classify the 2000 images sampled from the corresponding dataset into 5 categories. This fine-tuning was performed in the standard manner [64] by replacing the final (fully-connected) layer of the network to reflect the number of target classes in each dataset. The models learnt to minimise the cross-entropy error by using the Adam optimiser [65] with a mini-batch size of 32 and learning rate of $10^{-5}$, which was reduced by a factor of 10 on plateau using the Pytorch scheduler function `ReduceLROnPlateau`. In one simulation study (Fig 4), we used a network that was pre-trained on a variation of ImageNet that induces a shape bias [37] and then froze the weights in all but the final classification layer to ensure that the learned bias was present during the training on the new images. In all simulations, learning continued till the loss function had converged. Generally this meant that accuracy in the training set was > 99%, except in the case where we froze all convolution weights where accuracy converged to a value > 70%. Each model was tested on 500 images drawn from `Both`, `Conflict`, `Shape` and `Non-Shape` conditions outlined above. The results presented here are averaged over 10 random seed initialisations for each model. All simulations were performed using the `Pytorch` framework [66] and we used `torchvision` implementation of all models.

**Ideal inference model.** In order to understand how prior biases affect human and CNN classification in the new task environment, we compared their classification to a statistical inference model that computes the ideal category label for a stimulus based solely on the information observed in a sequence of trials. In all our experiments, a trial presented a mapping between a group of features and a category label. The goal of the Ideal Inference model was to accumulate this information over a sequence of trials to predict the mapping in a future trial. It does this by creating a generative model that predicts the probability of observing each feature, given a category label. For example, in Experiment 2, each trial presents a shape, five segment colours and a category label. Based on this information, we can update the generative model, assigning a higher probability for observing the shape and segments observed in the trial, given the class label. Over a sequence of trials, a participant will observe more colours, shapes and category labels and in each trial we can keep adjusting the generative model predicting shapes and colours given the class labels. In a test trial, we can then use the generative

model and Bayes' rule to infer the probability of all category labels given the observed shape and colours. We now describe this sequential Bayesian updating procedure formally.

The goal of this model is to answer the following question: what class, $Y \in \{1, \ldots, C\}$, should a decision-maker assign to a test image, given a set of mappings from images to class labels (training trials). For the purpose of statistical inference, each image can be treated as a vector of features and each training trial assigns a feature vector, $\mathbf{x_i} = (x_i^1, \ldots, x_i^F)$, to a class label, $Y = c$. In our behavioural task, each feature (colour / location / size) can take a discrete set of values, so we treat each feature as a categorical random variable, $X^f \in \{1, \ldots, K\}$. The decision-maker infers the class label for a test image, $\mathbf{x_{test}}$, in two steps. Like the neural network, it first learns a set of parameters $\boldsymbol{\theta}$ that encode the dependencies between class labels and feature values in the training data. It then uses these parameters to predict class label for a given test image, $\mathbf{x_{test}}$.

We start at the end. Our goal is to compute $p(Y = c | \mathbf{X} = \mathbf{x_{test}}, \mathcal{D})$, the probability distribution over class labels given the training data, $\mathcal{D}$, and a test image, $\mathbf{x_{test}}$. Using Bayes' law, we have:

$$p(Y = c | \mathbf{X} = \mathbf{x_{test}}, \mathcal{D}) \propto p(\mathbf{X} = \mathbf{x_{test}} | Y = c, \mathcal{D}) \; p(Y = c) \tag{1}$$

where $p(Y = c)$ is the class prior and $p(\mathbf{X} = \mathbf{x_{test}} | Y = c, \mathcal{D})$ is a joint class-conditional density—the probability of observing the set of features, $\mathbf{x_{test}}$, for a given class, $c$. In our behavioural tasks, each feature is independently sampled. This means that the joint distribution factorises as a product of class-conditional densities for each feature:

$$p(\mathbf{X} = \mathbf{x_{test}} | Y = c, \mathcal{D}) = \prod_{f=1}^{F} p(X^f = x_{test}^f | Y = c, \mathcal{D})$$

Our approach is to estimate these class-conditional densities by constructing a generative model $p(X^f = x_{test}^f | Y = c, \boldsymbol{\theta})$. Here $\boldsymbol{\theta}$ are the parameters of the model that need to be estimated based on training data. Since $X^f$ is a categorical variable, a suitable form for this parametric distribution is the multinomial distribution, $Mult(x_{test}^f | 1, \boldsymbol{\theta})$. The Bayesian method of estimating these parameters is to start with the prior distribution $p(\boldsymbol{\theta})$ and update it based on training data, $\mathcal{D}$, to obtain the posterior $p(\boldsymbol{\theta} | \mathcal{D})$. An appropriate prior for the multinomial is the Dirichlet distribution, $Dir(\boldsymbol{\theta} | \boldsymbol{\alpha})$, where $\boldsymbol{\alpha}$ are the hyper-parameters of the Dirichlet distribution. Here we assume the flat prior, $\boldsymbol{\alpha} = \mathbf{1}$, which corresponds to Laplace smoothing. For this Dirichlet-multinomial model, the update step involves counting the number of times each feature value occurs in the training data and adding these counts to the hyper-parameters [67].

Once we have a posterior distribution on the model parameters, $p(\boldsymbol{\theta} | \mathcal{D})$, we can obtain the required class-conditional densities, $p(X^f = x_{test}^f | Y = c, \mathcal{D})$ by integrating over these parameters. This leads to the following expression (see [68]):

$$p(X^f = x_{test}^f | Y = c, \mathcal{D}) = \frac{N_k + \alpha_k}{\sum_v N_v + \alpha_v}$$

Here $N_k$ is the number of times $X^f$ takes the value $k$ in the training data and the sum in the denominator is carried out over all possible values $\{1, \ldots, K\}$ of $X^f$. Thus this model predicts that the class-conditional density of observing a feature value during a test trial depends on the relative frequency with which the given feature value occurs during the training data. These class-conditional densities can be plugged back into Eq 1 to give the probability distribution over all classes given the test image, $\mathbf{x_{test}}$. In our Results, we report this probability for the labelled class averaged over all the test images in a test condition.

In all experiments, the class label, $Y$ can take one of five possible values, that is $Y \in \{1, \ldots, 5\}$. In Experiment 1, where the location and colour of a single patch is diagnostic, the feature vector on any trial, $\mathbf{x_{trial}}$, is $(x_{shape}, x_{loc}^1, \ldots, x_{loc}^F)$, where $x_{shape} \in \{1, \ldots, 5\}$ is a multinomial random variable for the shape feature that can take one of five values, and each of the $x_{loc}^f \in \{1, \ldots 20\}$ is multinomial random variable for a location that can take one of twenty possible colour values (we restricted the number of colours to 20 to make sure colours are clearly discernible by human participants). In Experiment 2, where the colour of one of the segments is diagnostic, the feature vector on a trial, $\mathbf{x_{trial}}$ is $(x_{shape}, x_{colour}^1, \ldots, x_{colour}^{20})$, where $x_{shape}$ is again a multinomial variable for the shape feature that can take one of five values and each $x_{colour}^f \in \{0, \ldots, 5\}$ is a count variables that represents the number of segments of colour, $f$, in the image. In Experiment 3, where the average size of patches is diagnostic, $\mathbf{x_{trial}} = (x_{shape}, x_{size})$, where $x_{size} \in \{1, \ldots, 5\}$ is a multinomial random variable for the average size of patches in the image. In Experiment 4, where the global colour of the figure is diagnostic, $\mathbf{x_{trial}} = (x_{shape}, x_{colour})$, where $x_{colour} \in \{1, \ldots, 5\}$ is a multinomial random variable representing the global colour of the figure.

## Supporting information

**S1 Movie. A short movie illustrating the task in Experiment 1.**
(GIF)

**S2 Movie. A short movie illustrating the task in Experiment 2.**
(GIF)

**S1 Fig. Results when both features are equally predictive.** Each panel shows the accuracy under the four test conditions for AlexNet (top row) or VGG-16 (bottom row). Each column corresponds to a different experiment. Both models were pre-trained on `ImageNet` and fine-tuned by reshaping the final layer to reflect the number of target classes in each experiment and trained on 2000 images from the training set (see Materials and methods for details). A comparison with Fig 3A shows that both architectures showed the same pattern of results as ResNet50: models were able to learn the task (high accuracy in the `Same` condition), learned both the `Shape` and `Non-shape` features (above chance accuracy in `Shape` and `Non-shape` conditions) and preferred to rely on the `Non-shape` feature (low accuracy in the `Conflict` condition).
(TIFF)

**S2 Fig. Results when non-shape feature is more predictive.** Each panel again shows the accuracy under the four test conditions for AlexNet (top row) or VGG-16 (bottom row). Each column corresponds to a different experiment. A comparison with Fig 3B shows that both architectures showed the same pattern of results as ResNet50: models showed a strong preference to rely on the non-shape feature in this case (a high-low-low-high pattern in the `Same-Conflict-Shape-Non-shape` conditions) and this preference became larger than the experiments where both features were equally predictive (compare with S1 Fig above).
(TIFF)

**S3 Fig. Results for learning without shape feature.** The two panels show accuracy in test blocks for AlexNet and VGG-16, respectively, when these models were trained on images that lack any coherent shape (Experiment 5). Each bar corresponds to the type of non-shape feature used in training. Like ResNet50, but unlike human participants (compare with Fig 5B), both models were able to learn all types of non-shape features.
(TIFF)

**S4 Fig. Two groups in Experiment 4a.** Each panel shows the accuracy under the four test conditions for a subgroup of participants. Participants were split based on whether they performed better in the shape or colour conditions. The first group contained N = 12 participants and the second group contained N = 13 participants.
(TIF)

**S5 Fig. Two groups in Experiment 4b.** Each panel again shows accuracy under the four test conditions for the subgroups of participants who prefer to rely on shape and colour, respectively. In this case, the first group consisted of N = 7 participants and the second group consisted of N = 18 participants.
(TIF)

**S6 Fig. Change in test performance with training in Experiments 1a, 2a, 3a, and 4a.** Fig 5A in the main text shows the change in performance under the four test conditions in Experiment 1b, 2b, 3b and 4b, where the non-shape feature and more predictive than shape features in training. Here we have plotted how performance changes in Experiments 1a, 2a, 3a and 4a, where both features are equally likely. Each panel shows how accuracy on the four types of test trials changes with experience. The top, middle and bottom row correspond to optimal decision model, CNN and human participants respectively. Columns correspond to different experiments. The scale on the x-axis represents the number of training trials in the top row, the number of training epochs in the middle row and the index of the test block in the bottom row. A comparison of S6 Fig and Fig 5A from the main text shows a very similar pattern in all experiments and for humans as well as the two types of models. The two models predict that a difference between `Both` and `Conflict` conditions emerges early and grows with learning. In contrast, human participants show no difference in the two conditions throughout the experiment in Experiments 1a, 2a and 3a. Further analysis of individual participants showed that, like Experiments 1b, 2b, 3b and 4b, no participant switched from using one feature to another during the experiment.
(TIFF)

**S7 Fig. Results for Experiments S1, S2 and S3.** In three experiments, we tested how participant behaviour changed when we presented the stimuli for a shorter duration (100ms) and restricted the field of view, such that the stimulus was always presented within 10° of fixation (see Materials and methods in main text). (a—S7A Fig) Accuracy of N = 25 participants in the four test conditions in an experiment that mirrors Experiment 1b—i.e., all training images contain a diagnostic patch and 80% images contain a diagnostic shape, (b—S7B Fig) Accuracy of N = 25 participants in the four test conditions in an experiment mirroring Experiment 2b—i.e., all training images contain a diagnostic segment and 80% images contain a diagnostic shape, (c—S7C Fig) Accuracy of two groups of N = 10 participants in test block where the training images contained only non-shape cues. Performance of participants in all experiments was consistent with their performance observed in other experiments. Though the overall accuracy of participants in this control experiments was slightly lower (mean accuracy in the `Both` condition was $M = 59.60\%$ in (a) and $M = 57.20\%$ in (b)), which is understandable given the faster presentation time, there was statistically no difference in their performance in the `Both`, `Conflict` and `Shape` conditions and their performance in the `Non-shape` condition was at chance. In Experiment (c), where there was no shape features in the training set, performance of both the `Patch` and `Segment` groups was statistically at chance. That is, participants consistently learned based on shape cues; when a diagnostic shape was not present during training, no participant managed to learn the task. (Compare results with Figs 3B and 5B).
(TIFF)

**S8 Fig. Results for Experiment S4.** Accuracy in the four conditions when participants are shown the stimuli for 3s instead of 1s. In this experiment, every trial has two diagnostic features—global shape and average size. Despite the increase in the duration of the stimulus, participants performed well in the `Both`, `Conflict` and `Shape` conditions, but performed at chance in the non-shape (`Size`) condition, indicating that they still preferred to learn based on shape. Notice, we used Experiment 3 (non-shape cue = average size) to test this because this is experiment in which the participants were most likely to pick on the non-shape (`Size`) cue based on results in Experiment 5, where mean performance in the Size condition was above chance, while mean performance in Segment or Patch conditions was at chance, even when there was no competing shape feature.
(TIFF)

**S9 Fig. Examples of stimuli in Experiment 1 (patch).** In each row we show (from left to right) an example image from the training set, `Both` condition, `Conflict` condition, `Shape` condition and `Non-shape (Patch)` condition for a category. Each image in the training set contains a diagnostic patch of a certain colour that is present at a category-specific location. Additionally, all training images in Experiment 1a and 80% of images in Experiment 1b have a diagnostic shape. Images in the `Both` condition contain both these features. Images in the `Conflict` condition contain the shape from one category but diagnostic patch from another category. Images in the `Shape` condition contain the shape feature but none of the diagnostic patches. Images in the `Patch` condition contain the diagnostic patch but none of the shapes from the training set.
(TIFF)

**S10 Fig. Examples of stimuli in Experiment 2 (segment).** In each row we show (from left to right) an example image from the training set, `Both` condition, `Conflict` condition, `Shape` condition and `Non-shape (Segment)` condition for a category. Each image in the training set contains a diagnostic segment of a category-specific colour. Only images of this category have a segment of this colour. Additionally, all training images in Experiment 2a and 80% of images in Experiment 2b have a diagnostic shape. Images in the `Both` condition contain both these features. Images in the `Conflict` condition contain the shape from one category but diagnostic segment from another category. Images in the `Shape` condition contain the shape feature but none of the diagnostic segments. Images in the `Segment` condition contain the diagnostic segment but none of the shapes from the training set.
(TIFF)

**S11 Fig. Examples of stimuli in Experiment 3 (size).** In each row we show (from left to right) an example image from the training set, `Both` condition, `Conflict` condition, `Shape` condition and `Non-shape (Size)` condition for a category. The average size of all images in the training set is diagnostic of the category. That is, different categories have images that have different average size of patches. Additionally, all training images in Experiment 3a and 80% of images in Experiment 3b have a diagnostic shape. Images in the `Both` condition contain both these features. Images in the `Conflict` condition contain the shape from one category but diagnostic size from another category. Images in the `Shape` condition contain the shape feature and the average size of patches is larger than the diagnostic size of any category in the training set. Finally, the `Size` condition contains images where the average size of patches is diagnostic but shape is not.
(TIFF)

**S12 Fig. Examples of stimuli in Experiment 4 (colour).** In each row we show (from left to right) an example image from the training set, `Both` condition, `Conflict` condition,

`Shape` condition and `Non-shape (Size)` condition for a category. All patches in an image have the same colour. This colour is diagnostic of an image's category in the training set. Additionally, all training images in Experiment 4a and 80% of images in Experiment 4b have a diagnostic shape. Images in the `Both` condition contain both these features. Images in the `Conflict` condition contain the shape from one category but diagnostic colour from another category. Images in the `Shape` condition contain the shape feature and a colour that is not diagnostic of any category in the training set. Finally, the `Colour` condition contains images with no coherent shape but where the colour of segments is diagnostic of the category. (TIFF)

**S13 Fig. Examples of stimuli in Experiment 5 and 6 (no shape).** Each row shows four examples from the training set that have the same category label as well as one example from the test set with the same label. The four rows correspond to the four conditions. In row 1, the predictive feature is patch location. In row 2, the predictive feature is colour of one of the segments. In row 3, the predictive feature is average size of patches. And in row 4, the predictive feature is colour of all patches. (TIF)

## Acknowledgments

We would like to really thank Casimir Ludwig, Laurence Aitchison and Milton Montero for their helpful advice and feedback during preparation of this manuscript.

## Author Contributions

**Conceptualization:** Gaurav Malhotra, Jeffrey S. Bowers.

**Data curation:** Marin Dujmović.

**Formal analysis:** Gaurav Malhotra, Marin Dujmović.

**Funding acquisition:** Jeffrey S. Bowers.

**Investigation:** Gaurav Malhotra, Marin Dujmović, Jeffrey S. Bowers.

**Methodology:** Gaurav Malhotra, Marin Dujmović.

**Project administration:** Jeffrey S. Bowers.

**Resources:** Jeffrey S. Bowers.

**Software:** Gaurav Malhotra.

**Supervision:** Gaurav Malhotra.

**Validation:** Gaurav Malhotra.

**Visualization:** Gaurav Malhotra.

**Writing – original draft:** Gaurav Malhotra.

**Writing – review & editing:** Gaurav Malhotra, Marin Dujmović, Jeffrey S. Bowers.

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
