## [Decision Letter · Decision Letter 0]

8 Dec 2021

Dear Dr. Malhotra,

Thank you very much for submitting your manuscript "Feature blindness: a challenge for understanding and modelling visual object recognition" (PCOMPBIOL-D-21-01905) for consideration at PLOS Computational Biology. As with all papers peer reviewed by the journal, your manuscript was reviewed by members of the editorial board and by several independent peer reviewers. Based on the reports, we regret to inform you that we will not be pursuing this manuscript for publication at PLOS Computational Biology.

Though the reviewers recognised that the experimental design was creative and appreciated the in-depth comparison between human and CNN behaviour, two of the three reviewers felt that this study does not help us discriminate between existing models, provide new paths forward for better models of human vision, nor provide sufficient new insights into human visual processing. In particular, it is already known that CNNs show differences to humans, e.g. they rely less on shape than humans, so that aspect of the results is a replication of this known phenomenon. A replication study itself would be fine, but that was not the goal here. This paper sought to make a stronger claim, namely that since humans tend to rely on global features like shape even when there is a stronger predictor in the data they were given in these experiments, then human vision cannot be based on statistical learning/inference. But, as the reviewers noted, this study does not actually demonstrate this. First, there are numerous differences between CNNs and humans (and their training) that could potentially explain the differences in performance on the tasks investigated here and which were by no means controlled for, and thus, we can't really say based on the data here what is causing the difference. Second, we cannot conclude from this experiment that humans do not rely on statistics for learning (nor that evolution didn't). For example, as Reviewer 3 noted, there is no way to simulate a real "zero experience" for humans given that the human visual system is pre-trained via evolution and development and humans have a lifetime of experience of interacting with objects in a 3D world that is reflected in their neural representations. As such, it could be that the amount or sort of data provided to the participants in this study was insufficient to overcome a strong bias towards global properties like shape that emerged from a lifetime (and evolutionary history) of such properties being statistically reliable predictors. Altogether, given these reviews and the considerations they raise, we unfortunately do not believe that this paper is suitable for publication at PLOS Computational Biology.

The reviews are attached below this email, and we hope you will find them helpful if you decide to revise the manuscript for submission elsewhere. We are sorry that we cannot be more positive on this occasion. We very much appreciate your wish to present your work in one of PLOS's Open Access publications. 

Thank you for your support, and we hope that you will consider PLOS Computational Biology for other submissions in the future.

Sincerely,

Blake A Richards

Associate Editor

PLOS Computational Biology

Samuel Gershman

Deputy Editor

PLOS Computational Biology

Reviewer's Responses to Questions

**Comments to the Authors: **

Reviewer #1: In this study, the authors explore the relative biases towards colour and shape in humans and deep convolutional neural networks using a supervised category learning task. They use some inventive new stimuli in which elementary colour patches are composed into segments, which are then assembled into an object.

The major findings are 1) that when global shape is predictive of the category, humans tend to ignore a colour cue that is either a single element or a segment of the object, but use the colour cue if it covers the entire object; 2) that CNNs, by contrast, prefer to use the colour patch, especially when it is made more diagnostic, by including 20% of shape “catch” trials [experimental series B]; 3) that pretraining on the style-transfer version of ImageNet developed by Geirhos et al increases the bias towards shape in the CNN, but not to human levels; that humans cannot use the colour patch or segment even if colour is non-diagnostic, and even if they are told about it.

I liked several things about this report. I thought the stimuli were creative and had the potential to be useful tools for future experimentation. I appreciate the in-depth attempt to compare humans and CNNs on a novel task. The study is elegantly executed and reported. I like the factorial design in which cues were either presented singly or together (coherently or in conflict). The writing is for the most part clear (except for some omitted/buried details about the task).

The question I asked myself whilst reading was what exactly we learned from the results that was new. It is well known in the category learning literature that 1) humans exhibit a shape bias, and 2) they struggle to solve high-dimensional classification tasks with novel stimuli and no curriculum. For example, for the latter see the Demons experiments in Pashler & Mozer 2013; there are numerous other examples. It’s also been shown a number of times that when multiple cues predict a category label, humans can easily overlook one, even if it is quite salient. Schuck (2015, Neuron) is one example. The human findings here seem to replicate these results, albeit with a new task and stimuli. They do so elegantly, but I found it hard to be surprised by any of the human results that the authors report.

The comparison to CNNs is instructive, but also follows a tradition which has shown that CNNs struggle to recognise global ensembles but are good at using local diagnostic elements (or textures) to solve classification problems. This is perhaps not surprising when you think about their architecture. The convolutional filters learned at each local image location are shared automatically across visual space. This means that they are given a very strong inductive bias to learn position-invariant local cues. By contrast, shape and other global features of an array have to be learned in the final layers where disparate interactions across the entire image can be computed. CNNs are built with the wrong inductive bias to recognise shapes. Papers by Bob Geirhos and Sam Ritter, which the authors cite, have made this point before.

I also had trouble understanding many aspects of the report, at least at first. Perhaps these could be clarified for other readers.

- I initially assumed it was a binary classification task. It would be good to state that it is a 5-way classification task up front, rather than expecting the reading to dig it out of the methods.

- Were the training trials identical to the test trials (except for the provision of supervision?). 

- It would be good to know more details about the constraints on stimulus generation. The authors don’t say how the different shapes were generated. How were they different from one another?

- it would be good to know exactly what generalisation is over. Was it just position and rotation of the shape that varied? 

- the authors say that the "location" of the colour patch varied. by location relative to what? presumably not in native space, becuase the shapes change position. relative to the segment? or the entire shape?

- more details are needed about the probabilistic model. How was its state space constructed? I couldn’t follow what exactly was done. In general, it might be useful to better motivate this model, which is not really discussed in detail.

Reviewer #2: Summary

This article compares human categorization performance to a convolution neural network (CNN) and a Bayesian statistical model (Ideal Classifier; IC). The stimuli for classification were novel, colored shapes. 

In experiments 1-4, classification (into 1 of 5 categories) was possible through a conjunction of virtual object shape information and one of (a) patch location (a part of the virtual object appearing in a particular location and with a particular color), (b) segment color (part of the virtual object having a particular color), (c) average size of the virtual object, and (d) the color of the entire virtual object. Each experiment was paired with a second part, in which classification, during training, by shape information was only possible on 80% of trials, while the other cue to classification remained 100% predictive.

Classification accuracy following training was measured in 4 conditions (a) both cues valid, (b) shape information valid, but the non-shape cue in conflict (indicating a different category), (c) the shape cue valid and the non-shape cue non-informative, and (d) the shape cue non-informative and the non-shape cue valid. The pattern of results for human participants suggested a strong bias to categorize based on shape, even when, during training, shape was less predictive of category membership than the non-shape cue. Use of the non-shape cue was negligible, except when the non-shape cue was the color of the entire virtual object. Differently, the CNN performed in a manner indicating a bias to classify based on the non-shape cue (but see below). The IC generally performs as expected - classifying at near ceiling in all cases except when the non-shape cue is in conflict.

A CNN trained to have a shape bias classified in a similar manner, but with more sensitivity to shape information. This result remained, even when pre-classification layers of the network were frozen, presumably maintaining much of the pre-trained shape bias.

In a fifth experiment, the same general procedure was used, but now with the absence of shape as a cue to category membership. Here as well, participants failed to learn to categorize based on patch location or segment color, but did learn to categorize based on size and color. The CNN, predictably, learned to categorize in all of these cases.

A sixth experiment showed that, even when participants were instructed to use the non-shape cue, they still struggled to do so, supporting the hypothesis that limited cognitive resources underlies the bias to rely on global shape information, even when this strategy is suboptimal.

Review

The impressive performance of CNNs, as compared to humans, have led some researchers to claim that the underlying representations, in network and in human, may be similar. However, the sometimes strange strategies employed by CNNs (e.g, using background information to classify an object) suggest fundamental differences from humans. This paper investigates if, how, and why CNNs might have different classification strategies. The results are, to me, interesting and convincing. CNNs are biased toward classifying virtual objects based on local features, whereas humans are biased to classify virtual objects based on global shape. The authors show that these biases are robust and fit with a theory they advance (and some modest evidence they present) that the human strategy results from cognitive limitations that are (relatively) absent for the CNN.

In general, I think the paper can be accepted with minor revisions mostly related to clarification:

1. It is unclear how the Conflict test trials are scored (e.g., in Figure 2) when the concept is introduced on page 5-6.

2. Some more care could be taken to describe the four test trial types more clearly. Nothing in the paper is wrong, but describing what each is (and is not), placed in list form outside the flow of the text would be helpful.

3. In Figure 1 caption, “movies S2 and S2” appears to be a typo.

4. It is unclear to me why the performance of the IC is so low for conflict cases in Experiment 1. Wouldn’t the statistical properties of the training and testing sets, wrt classification, be the same as in Exps. 2, 3, and 4?

5. It seems a priori obvious that humans would not learn the patch location cue without enormous amounts of training, and maybe not even then. It’s not terribly surprising that, even when told the diagnostic cue, performance in the patch condition was at chance. What is surprising, however, is that performance in the segment condition was also at chance, even when the relationship between the cue and the categories was made apparent to the participants. Is there a better explanation for why participants were at chance in this condition, other than that they didn’t understand what they were supposed to do? Given the instructions, as described in the procedure section, it’s hard to understand why participants were at chance for this condition in Experiment 6. 

6. Swap is used instead of Conflict in some appendix figure captions

7. A diagram of the experimental procedure (for humans) is always, IMO, helpful.

Reviewer #3: In this study, Malhotra et al., primarily probed the differences between CNNs and humans based on behavioral comparisons done with categorization tasks on images that were synthesized by varying specific image features. They conclude that these results highlight a fundamental difference between statistical inference models and humans. 

Although the authors perform multiple experiments and several comparisons between CNN behavior and human behavior, it is very unclear to me what the central claim (or novel result) of the paper really is. Is it mainly that current CNNs are not fully predictive of human behavior (under the current specific training diet, architectural choices, learning objectives etc.)? If so, that point has already been made in multiple studies — and also not quantitatively assessed (with appropriate noise ceilings) in this study. Are the measurements done in this study better to falsify certain models or discriminate among similarly performing models than those done in earlier studies? It is not clear. The experiments (in this study), in my view were done (and presented) as to provide some insights into why such differences (between CNNs and humans) might exist. I have listed my major concerns with that approach below. 

In line 68, the authors explicitly state that “Our objective was to see whether human inferences match the statistical inferences predicted by the ideal inference model and the CNN”. 

First, to make these comparisons, certain very basic premises need to be set. Other than differences in training of CNNs vs humans, it is critical in studies where CNNs and humans are compared to make specific (well grounded) commitments as to exactly how the CNNs are treated as primate vision models. What is the size of the input stimuli (that is field of view)? How long are the stimuli shown (i.e., is it rapid, “one-pass” for humans like in CNNs or long enough to engage recurrence)? Which layer is mapped onto to what brain area or what assumptions are made for the behavioral readout algorithms? All of these have been carefully explored in many different studies. Without establishing some very basic commitments, it is hard to evaluate the results of such studies with respect to the larger state of the field. Please see details below:

Field of view: What are the size of the presented images in visual degrees (for humans)? The authors state, “For the behavioural experiments, the stimuli size was scaled to 90% of the screen height (e.g. if the screen resolution was 1920x1080 the image size would have been 972x972). This ensured that participants could clearly discern the smallest feature in an image (a single patch) which we confirmed in a pilot study”. That way of defining image size is not consistent with other studies and with usual regimes of images sizes (8 to 10 visual degrees) where the CNNs best predict human behavior. 

Presentation time: Most experiments in this study were run with 1000 ms to 3000 ms image presentations. These are timescales when various recurrent computations come into play (that are missing in current CNNs, shown across multiple research papers). The current CNNs are best approximations of early responses of the brain and the behavior that is also tested at those timescales. 

Layer mapping: The fine-tuning of the model was done based on the penultimate layer of the model. It is not always the case that these are the best layers to predict human behavior (or inferior temporal cortex representations). In the Discussion, the authors mention: “These results pose a challenge for the hypothesis that humans and CNNs have similar internal representations of visual objects.” This was a purely behavioral study and no concrete claims about any match between “internal representations” of CNNs and humans should be made. There were no test/comparison of internal representations in this study. So such conclusions are not supported by data from this study. From this study, only a behavioral comparison can be made between these two systems (CNNs and humans).

Second, this specific question (do CNNs and humans make different inferences) has been asked multiple times before. It has been repeatedly tested before (in other studies) and most (if not all) models have been already falsified under various metrics (the authors cite some of those studies). Therefore an additional value that such a study can bring is if they provide a precise strategy (a training image-set, constraining data, or specific architectural insights) to develop better models of human vision or provide computationally explicit guidance. Unfortunately, in my opinion, the current study provides neither of these in a quantifiable or model-implementable way. 

The authors make the claim that differences between CNN and humans may be due differences in training environment. They claim that “To overcome these limitations, we conducted a series of experiments and simulations where humans and models had no prior experience with the stimuli.” --- There is no way to simulate a real "zero experience" for humans given that the human visual system is trained via evolution and development. Humans (as a species) likely have more than a lifetime of experience with objects that is reflected in their neural representations. They do not in all likelihood start with a blank slate. It is very difficult to make that assumption for humans, and the controls that the authors have done are certainly not complete. It is for the same reason that most studies only test the inference capability of trained CNNs (a “learned system” ) when it comes to comparing CNNs with humans. 

The authors also claim that, “participant behaviour is better explained by a “satisficing” account [53] than an optimising account of object recognition”. 

And, “Instead of an optimisation approach that underlies many machine learning models, we argue that human behaviour is much more in line with a satisficing account [53], where features are selected because they allow participants to perform reasonably on the task while taking into account their limited cognitive resources. While performing statistical inferences is certainly important, models of vision must also consider the cognitive costs and biases in order to be realistic theories of human object recognition.”

How do they measure the “limited cognitive resources”? Can that be tested in models or clearly defined and measured in humans for this study and the specific comparisons made? These words need to be operationalized as clear, falsifiable model components if they are used as speculations for mechanisms. Also, other words like “computational cost”, “cognitive capacities” etc. have been used for CNNs and humans without any concrete measurement or operationalization for most (if any) of them. 

Some more specific comments:

51— Authors claim “while both system have the same learning objective” — that is a hypothesis/speculation and not a fact. So it should be written as such. 

220- “If CNNs and humans are driven by performing statistical inferences, we expect both systems to start relying on the feature that is better at predicting the category label.” — This is a product of merging the learning and inference ideas. You can have two models where the internal representations can be the product of optimization on a very specific objective. But now when you take these representation to do another specific task — you can readout the behavior in two completely different ways (i.e., two different decoding models on same feature sets) to come up with massive mismatches in their output performance patterns. What the authors are trying to claim, cannot be tested unless they are really comparing the internal representations of CNNs with some neural data from humans. 

605 - “Since participants had a lifetime experience of classifying naturalistic objects prior to the experiment, we pre-trained our networks on a set of naturalistic images (ImageNet).” — This pre-training can also be assumed as the simulation of evolution instead of learning during the lifetime. The fine-tuning that follows after this — is a more likely substitute for learning during lifetime.

**Have the authors made all data and (if applicable) computational code underlying the findings in their manuscript fully available?**

Reviewer #1: No: 

Reviewer #2: Yes

Reviewer #3: Yes

PLOS authors have the option to publish the peer review history of their article (what does this mean?). If published, this will include your full peer review and any attached files.

Reviewer #1: No

Reviewer #2: No

Reviewer #3: No

---

## [Decision Letter · Decision Letter 1]

19 Mar 2022

Dear Dr. Malhotra,

We are pleased to inform you that your manuscript 'Feature blindness: a challenge for understanding and modelling visual object recognition' has been provisionally accepted for publication in PLOS Computational Biology.

Best regards,

Blake A Richards

Associate Editor

PLOS Computational Biology

Samuel Gershman

Deputy Editor

PLOS Computational Biology

Reviewer's Responses to Questions

**Comments to the Authors:**

Reviewer #2: The authors have addressed all my concerns. I'll leave it to the other reviewers to decide if their concerns, which I agree have merit, have been sufficiently addressed.

Reviewer #3: I thank the authors for making significant revisions to their original manuscript, running new experiments, and addressing most of my comments. Some of the comments were not fully addressed given that those would require experimental work (out of scope for this project). Therefore, I would recommend publication.

**Have the authors made all data and (if applicable) computational code underlying the findings in their manuscript fully available?**

Reviewer #1: Yes

Reviewer #2: Yes

Reviewer #3: None

PLOS authors have the option to publish the peer review history of their article (what does this mean?). If published, this will include your full peer review and any attached files.

Reviewer #1: No

Reviewer #2: No

Reviewer #3: No

---

## [Editor Report · Acceptance letter]

24 Apr 2022

PCOMPBIOL-D-21-01905R1 

Feature blindness: a challenge for understanding and modelling visual object recognition

Dear Dr Malhotra,

I am pleased to inform you that your manuscript has been formally accepted for publication in PLOS Computational Biology. Your manuscript is now with our production department and you will be notified of the publication date in due course.

With kind regards,

Agnes Pap
